Access to scientific literature by the conservation community

Larios Daisy daisy.larios@iucn.org 1
Brooks Thomas M. 1
Macfarlane Nicholas B.W. 2
Roy Sugoto 3
1 Science and Knowledge Unit, International Union for Conservation of Nature (IUCN) , Gland , Vaud , Switzerland
2 Science and Knowledge Unit, International Union for Conservation of Nature (IUCN) , Washington D.C. , United States of America
3 Global Species & Key Biodiversity Areas Programme, International Union for Conservation of Nature (IUCN) , Gland , Vaud , Switzerland
Lambert Max
Electronic publication date: 2020 Jul 9
Publication date: 2020
Volume: 8
Electronic Location ID: e9404
Received 2019 Dec 17; Accepted 2020 Jun 1
Copyright: ©2020 Larios et al.
Copyright year: 2020
Copyright holder: Larios et al.
License: This is an open access article distributed under the terms of the Creative Commons Attribution License, which permits unrestricted use, distribution, reproduction and adaptation in any medium and for any purpose provided that it is properly attributed. For attribution, the original author(s), title, publication source (PeerJ) and either DOI or URL of the article must be cited.
License URL: https://creativecommons.org/licenses/by/4.0/

Keywords: Information seeking, Libraries, Open access, Access to literature, Biodiversity conservation, Conservation organisations

Funding: The authors received no funding for this work.

==============================
Access to the scientific literature is perceived to be a challenge to the biodiversity conservation community, but actual level of literature access relative to needs has never been assessed globally. We examined this question by surveying the constituency of the International Union for Conservation of Nature (IUCN) as a proxy for the conservation community, generating 2,285 responses. Of these respondents, ∼97% need to use the scientific literature in order to support their IUCN-related conservation work, with ∼50% needing to do so at least once per week. The crux of the survey revolved around the question, “How easy is it for you currently to obtain the scientific literature you need to carry out your IUCN-related work?” and revealed that roughly half (49%) of the respondents find it not easy or not at all easy to access scientific literature. We fitted a binary logistic regression model to explore factors predicting ease of literature access. Whether the respondent had institutional literature access (55% do) is the strongest predictor, with region (Western Europe, the United States, Canada, Australia and New Zealand) and sex (male) also significant predictors. Approximately 60% of respondents from Western Europe, the United States, Canada, Australia and New Zealand have institutional access compared to ∼50% in Asia and Latin America, and ∼40% in Eastern Europe and in Africa. Nevertheless, accessing free online material is a popular means of accessing literature for both those with and without institutional access. The four journals most frequently mentioned when asked which journal access would deliver the greatest improvements to the respondent’s IUCN-related work were Conservation Biology, Biological Conservation, Nature, and Science. The majority prefer to read journal articles on screen but books in hard copy. Overall, it is apparent that access to the literature is a challenge facing roughly half of the conservation community worldwide.

Introduction

A commonly held belief through the conservation community is that lack of access to the scientific literature is a limiting factor for practitioners (Fonseca & Benson, 2003; Rafidimanantsoa et al., 2018; Amano, Gonzalez-Varo & Sutherland, 2016). This assumption stands to reason given the evidence that access to information would improve conservation outcomes (Cook, Hockings & Carter, 2010; Walsh, Dicks & Sutherland, 2015) as well as the documentation of shortfalls in literature access from other fields of applied science (Horton, 2000; Godlee et al., 2004). This creates a challenge for conservation, especially given that there appears to be an inverse relationship between where research takes place and where it is most needed (Rodrigues et al., 2010; Wilson et al., 2016). Meanwhile, library science literature has generally focused its studies on the information needs and behaviours of scientists and scholars, only more recently expanding its scope to consider the needs of nonacademic professionals (Leckie, Pettigrew & Sylvain, 1996). For conservation, previous studies have found that those in sectors other than academia and government experience the most difficulty in finding the biodiversity information they need to do their work (Steiner Davis et al., 2014; Fabian et al., 2019). Despite some evidence that scientific journals do not contain the type of information considered most important by conservation professionals (Roy, Smith & Russell, 2009; Fabian et al., 2019), the degree to which access to the scientific literature meets the stated needs of the global community has never been assessed, and little consideration has been given to the role of libraries in facilitating access to literature.

Existing models of information seeking tend to focus on specific professionals or academic groups, but biodiversity conservation is undertaken by a web of actors that goes beyond scientists and academics to include on-the-ground practitioners as well as employees of NGOs and governments. We therefore surveyed the constituency of the International Union for the Conservation of Nature (IUCN) to determine the extent of literature access from among the world’s conservation professionals and to which their institution facilitates access to literature.

Created in 1948, IUCN is a Membership Union uniquely composed of both governments and state agencies (223 in total) and civil society and indigenous peoples’ organisations (1,117 in total), with each of these two houses having equal weight in the Union’s governance. Members approve the mandates of expert Commissions, of which there are currently six, encompassing some 13,000 experts who lend their expertise to IUCN. The Members also elect a Council that appoints a Director-General, who in turn recruits a professional Secretariat, comprising roughly 1,000 employees. Given this breadth of IUCN’s makeup, respondents to our survey could have a variety of backgrounds and roles: from environmental practitioners, nonprofit workers, and governmental decision makers to academics and consultants. Here, we refer to this complex group of survey respondents as “conservation professionals” for simplicity’s sake. Nonetheless, it could include respondents who work within environmental organisations in financial, administrative, or legal capacities (i.e., employees of IUCN Members or the IUCN Secretariat) and exclude conservation professionals who are not IUCN Commission members and if they work for organisations whose focus is not conservation (e.g., watershed councils and city governments) or that otherwise are not Members of IUCN.

IUCN has always served a role in supporting access to conservation knowledge and literature, a role historically held to be critical to supporting the goals of conservation. When it was founded in Fontainebleau on 5 October 1948 as the International Union for the Protection of Nature (IUPN), one of its original objectives was to “collect, analyse, interpret and disseminate information about the ‘Protection of Nature”’ (Büttikofer, 1946). It regarded the International Office for the Protection of Nature, one of its founding international organisational members, as essential in carrying out this objective (IUCN, 1951). The Office’s predecessor, the Central Bureau of Information and Correlation, was founded at the 1928 General Assembly of the International Union of Biological Sciences by the National Committees of Belgium, France and Holland, who saw the Bureau as an important step towards the ultimate goal of creating an international union (Büttikofer, 1947). Organisations dedicated to the protection of nature were to send publications to the Bureau to facilitate the later establishment of this international union (Büttikofer, 1946). The Bureau was replaced by the International Office for the Protection of Nature in December 1935 and transferred to Amsterdam in 1940 at the outbreak of World War II, which severely limited the Office’s finances. By 1947, though, it had been modestly reestablished as a “scientific institution, a library, a record-office, a centre for receiving, classifying and publishing data, for organising inquiries, for propaganda and information” (Büttikofer, 1947). The Office finally merged with IUPN in 1955, taking the name of the Office’s founder: Bibliothèque van Tienhoven. The IUCN HQ Library over the years has built upon the original collection inherited from the Office.

We intend for the results of this survey to have immediate practical implications. Most directly, our results will steer the strategy for IUCN and other conservation organisations in strengthening their institutional commitment to their own libraries. Second, they should also provide useful insight for conservation libraries housed throughout the IUCN Membership. Equally, actors in the complex publishing landscape of conservation research—involving commercial publishers, non-profit publishers, universities, academics and conservation organisations under a number of arrangements—may be able to draw from our findings to enhance their readerships and impact. Finally, our results may be valuable to foundations and other funding agencies that support conservation, in seeking to optimise their investments.

Materials and Methods

The survey consisted of fifteen questions divided over four pages (Supplemental Information). References to “scientific literature” throughout the survey were defined as “peer-reviewed scientific journals plus technical books” in the introductory text to the survey. We define “institutional access to scientific literature online” to mean that the respondent’s employer or some other institution to which they have an affiliation (e.g., a university) has a library or library-like department that negotiates online subscriptions to journals or databases on behalf of the institution’s users. We did not use the word “library” because users may strictly associate libraries with a physical space, unaware that access to journals or databases—often seamlessly authorized by IP address—is facilitated by the institution’s library (Tenopir, Christian & Kaufamn, 2019). A library does not necessarily have to be in a physical space, as can be seen in the definition proposed by the American Library Association : “A library is a collection of resources in a variety of formats that is (1) organized by information professionals or other experts who (2) provide convenient physical, digital, bibliographic, or intellectual access and (3) offer targeted services and programs (4) with the mission of educating, informing, or entertaining a variety of audiences (5) and the goal of stimulating individual learning and advancing society as a whole” (American Library Association, 2020). We made reference throughout the survey to access to scientific literature for the purposes of “IUCN-related work”, given the scope of the IUCN HQ Library.

The survey’s first page collected demographic information about the respondent, with a fourth question asking how frequently the respondent perceived that they should be consulting scientific literature to carry out their IUCN-related work. We utilized the word “should” to distinguish between actual and required use of literature, since actual use could be suppressed due to lack of access. Results for the remaining questions only include those of respondents who required scientific literature in the course of their IUCN-related work; those who answered “Never” to this question were taken to the final page of the survey. The second page used multiple-choice questions to determine the ease and importance of the respondent’s access to the literature; asked which one journal would have the largest impact on the respondent’s work were they to have access; and explored preferred reading formats, whether the respondent has institutional access to the literature, and frequency of different methods of literature access. The survey logic was designed so that those who reported no institutional access were taken to a third page, which asked respondents to assess likely frequency and impact of use were they to have such access. The final page offered respondents the opportunity to leave comments and contact details.

The survey was made available in all three official IUCN languages (Spanish, French, English) via two separate emails on 19 July 2016 to (i) primary contacts for all IUCN Member organisations, who were asked to forward the message to those individuals undertaking IUCN-related work within their institution, (ii) all IUCN Secretariat staff, and (iii) all members of the six IUCN Commissions for 2013–2016. These categories are non-exclusive: an individual could be a member of more than one Commission, or could simultaneously be a Commission member and an employee of a Member organisation or of the IUCN Secretariat. Membership sizes of the Commissions vary, with most having ∼1,000 members and the Species Survival Commission having ∼10,000 members. The language in which the survey was sent was determined by whether the contact had an indicated language of preference in IUCN’s customer relationship management (CRM) system; those without a preference received the English-language version by default. We sought to be inclusive of all who had any need for scientific literature in their IUCN-related work and did not seek to limit the survey to those of particular roles or backgrounds. Therefore, our survey results likely include some responses from individuals who work in areas other than biodiversity conservation and require other types of literature e.g., legal or management literature. We sent a reminder on 10 August 2016 and the survey was closed on 12 August 2016. The survey was wholly voluntary.

We aggregated results by country according to the UN regional groups—Africa, Asia-Pacific, Eastern Europe, Latin America and the Caribbean, and Western Europe and Others (which includes the United States, Canada, Australia and New Zealand). While a range of other national socio-economic parameters (e.g., GDP, income equality, education of girls and boys) could be included, we chose to select these regional groupings to reflect political and social as well as economic similarities in as small a number of groups as possible, in a way informative for decision-making in conservation, libraries, and other relevant institutions.

To compare the relationship between a respondent’s answers to the demographic and professional questions and their perception of ease of access to necessary literature, we modeled ease of access by condensing responses to the ease of access question into a binary variable (very hard + hard = 0, easy + very easy = 1) and fitting a binary logistic regression model to the full rank dataset of 1,970 respondents who answered all questions under consideration in the model. We began with consideration of five variables suspected likely to influence ease of literature access: institutional access (yes/no), institutional affiliation (five categories), discipline as reflected by Commission membership (six non-exclusive categories], sex (two categories), and region (five categories). Language was not included as a factor given the relatively low number of responses in Spanish and French compared to English; however, responses from all three language variations of the survey were included in the model. Standard variable selection approaches based on AIC scores (Akaike, 1974), resulted in a final model of the probability of access being easy as a function of Region (as compared to a base case region of Africa), Sex (“male” compared to “female”), and Institutional Access (“yes” compared to “no”) (Table 1). The base case of Africa, female, and no institutional access was chosen for comparison because those respondents reported the most difficult access. Institutional affiliation and Commission membership did not emerge as significant predictors in the model. In addition, interactions were explored between sex and region, and sex and institutional access, neither of which were significant. There was some evidence for an interaction between institutional access and being in the Western Europe and Others Group, which did not change the overall conclusions and was not included given the principles of parsimony and statistical efficiency, and the complexities of interpreting interaction terms in non-linear models (Ai & Norton, 2003). All model fitting was conducted using R (R Core Team, 2017).

Results

In total, we received 2,285 responses to our survey, though not all respondents answered all questions. This represents 13% of the IUCN constituency to whom the survey was directly distributed, although it is difficult to give a precise return rate given that there would have been some overlap between contact lists and as the actual number of potential participants among IUCN Members is unknown (Table 2). Anecdotal email responses suggest that some Member focal points erroneously thought the survey should be filled out on behalf of the entire organisation. Also, our results will be biased against those who did not have internet access during the time of survey (who are in turn likely to have poor access to the scientific literature in the first place). Nearly all (87%) responses were to the English-language version, and the vast majority (97%) of respondents felt they should be accessing scientific literature at least once per month (Fig. 1).

Table 1 Summary of the final binomial logistic regression model.

Coef. shows the change in the log-odds for a change between two cases of the same variable since they are all categorical (eg. moving female to male). In the case of region, the comparison is between each region code and Africa, since that region has the lowest log odds of easy access. The comparison reference level (Africa, Female, no Institutional Access) is not shown. Odds ra. is the odds ratio. Std. Err. is the standard error of the estimate, z and p are the Wald z-statistic and associated p-values.

Ease of Access	Coef.	Odds ra.	Std. Err.	z	p	
Model Intercept	−1.5754	.2069	0.1711	−9.207	<2e−16***	
Region: Asia-Pacific	0.2610	1.2982	0.1819	1.435	0.151	
Region: Eastern Europe	0.2266	1.2544	0.2729	0.831	0.406	
Region: Latin America and Caribbean	0.1824	1.2001	0.1940	0.940	0.345	
Region: Western Europe and Others	0.5467	1.7275	0.1587	3.445	0.000572***	
Sex: Male	0.3208	1.3782	0.1117	2.873	0.00407***	
Institutional Access: Yes	1.9251	6.8558	0.1028	18.723	<2e−16***	
Notes.

Bold styling indicated rows that were significant at a p value of less than 0.05.

Table 2 Survey response rate among all to whom the survey was directly distributed, by IUCN component.

Respondent type	Number of responses	Total sent	Response rate	
All respondents	2,285	17,166	13.31%	
Any Member	504	1,609	31.32%	
State Member	68	Unknown	Unknown	
IUCN Government Agency Member	78	Unknown	Unknown	
IUCN International NGO Member	124	Unknown	Unknown	
IUCN National NGO Member	237	Unknown	Unknown	
IUCN Affiliate Member	53	Unknown	Unknown	
IUCN CEC member	155	1,152	13.45%	
IUCN CEESP member	127	578	21.97%	
IUCN CEM member	165	968	17.05%	
IUCN SSC member	1,050	9,528	11.02%	
IUCN WCEL member	77	1,377	5.59%	
IUCN WCPA member	407	2,594	15.69%	
IUCN Secretariat staff	155	1,057	14.66%	

Figure 1 Required frequency of access to scientific literature (n = 2, 285).

How easily can the conservation community access scientific literature?

The survey revolved around the question, “How easy is it for you currently to obtain the scientific literature you need to carry out your IUCN-related work?” Roughly half (49%) of all 2,004 respondents to this question find it not easy or not at all easy to access scientific literature (Fig. 2).

Figure 2 Ease of access to scientific literature in the conservation community (n = 2, 004).

Overall, 47% of the 2,004 respondents to the question reported having no institutional access to scientific literature online, which correlates greatly to ease of access to literature. Among those with online institutional access, 72% found it easy to obtain access to required literature. By contrast, a similar percentage (74%) of those reporting no institutional access found it difficult to access scientific literature (Fig. 3).

Figure 3 Ease of access to scientific literature among those in the conservation community according to whether they reported having institutional access to scientific literature (n = 2, 004).

Not surprisingly, then, institutional access was the primary explanatory variable predicting ease of access. Exponentiating the model coefficient shows that institutional access increased the odds of easy access to literature by a factor of 6.86; it would seem that affiliation with an institution with a library greatly increases the odds of easy access to scientific literature. Being male and being based in the Western Europe and Others Group were also significant predictors of ease of access (Table 1).

Respondents to our survey were based in 170 countries, allowing us to examine variation across the five United Nations regional socio-geographical groupings. Nearly half of respondents belonged to the Western Europe and Others Group (Fig. 4). The two socio-geographic areas with the greatest difficulty in obtaining scientific literature were Africa and Eastern Europe, with 63% of respondents from Africa and 57% of respondents from Eastern Europe reporting that accessing scientific literature as not easy or not at all easy (Fig. 5). Not surprisingly, these two regions also reported the least online institutional access to scientific literature (Fig. 6). This supports our model findings that being based in a country in the Western Europe and Others group as opposed to one in Africa increased the odds of easy access by a factor of 1.73, as shown by exponentiating the region coefficient (Table 2). Other regions were not significant predictors. A Tukey’s post hoc test showed regional differences between Africa and Western Europe and Others (p = 0.005), but no significant differences between all other pairwise combinations of regions (Table 3).

Figure 4 Survey respondents grouped by region (n = 2, 254).

Figure 5 Levels of ease of access to scientific literature for IUCN-related work among respondents from the five UN regions (n = 1, 982).

Figure 6 Levels of reported online institutional access to scientific literature among respondents from the five UN regions (n = 1, 982).

Table 3 Tukey post hoc contrasts between regions.

Contrast	Est.	Std.Err.	z	p	
Asia-Pacific - Africa	0.2610	0.1819	1.4350	0.5891	
Eastern Europe - Africa	0.2267	0.2729	0.8310	0.9161	
Latin America and Caribbean - Africa	0.1824	0.1940	0.9400	0.8739	
Western Europe and Others - Africa	0.5467	0.1587	3.4450	0.0048	
Eastern Europe - Asia Pacific	−0.0344	0.2620	−0.1310	0.9999	
Latin America and Caribbean - Asia-Pacific	−0.0786	0.1781	−0.4410	0.9915	
Western Europe and Others - Asia Pacific	0.2857	0.1388	2.0580	0.2259	
Latin America and Caribbean - Eastern Europe	−0.0442	0.2698	−0.1640	0.9998	
Western Europe and Others - Eastern Europe	0.3200	0.2458	1.3020	0.6759	
Western Europe and Others - Latin America and Caribbean	0.3643	0.1519	2.3990	0.1081	
Notes.

Bold styling indicated rows that were significant at a p value of less than 0.05.

More than twice as many men (1,556 respondents) as women (710 respondents) took the survey. Of the 604 female respondents to the question about institutional access, 52% reported having institutional access, compared to 54% of the 1,387 male respondents to this question. When all other factors were held constant, our final model predicts that men have higher odds of easy access than women, at an odds ratio of 1.38 (Table 2). Interactions between gender, region, and institutional access were not significant, so there is not strong evidence for co-variation between gender and other variables in the model. However, the number of male and female respondents could potentially impact the interpretation of the sex effect if they don’t appropriately reflect the population.

Overall, 1,738 of our survey respondents reported being a member of one (or more) of IUCN’s six expert Commissions. By taking Commission membership as a proxy for discipline specialisation, we examined variation across thematic issues in conservation. (This approach excludes the 547 respondents who do not belong to any Commission). Numbers of responses mirrored the size of each of the six Commissions. Overall, membership in a particular Commission did not emerge as a significant predictor of ease of access in our model. Institutional access to the scientific literature did vary though, from 60% among those whose specialisation includes environmental law to 42% among those whose expertise includes protected areas (Table 4).

Table 4 Disciplinary variation in proportion of respondents with institutional literature access.

IUCN Commission	Disciplinary specialisation	Responses to Q9 (number)	Institutional access (percentage)	
Commission on Education and Communication (CEC)	Environmental education and communication	125	52	
Commission on Environmental, Economic, and Social Policy (CEESP)	Environmental social science	112	54	
Commission on Ecosystems Management (CEM)	Ecosystem conservation	156	54	
World Commission on Environmental Law (WCEL)	Environmental law	58	60	
World Commission on Protected Areas (WCPA)	Protected areas	370	42	
Species Survival Commission (SSC)	Species conservation	950	58	

Overall, 656 of all survey respondents reported being an employee of IUCN itself or an IUCN Member organisation, which we used to assess variation by sector. However, as respondents as a whole were not specifically asked to identify their work sector or employer, this partial snapshot excludes the work sectors of the 1,524 respondents who identified solely as Commission members. Sector categories are non-mutually exclusive, as 34 respondents selected more than one Membership category (presumably these are individuals who have multiple institutional affiliations). Here we consider responses from the IUCN Secretariat as well as four of IUCN’s Membership categories, combining responses from staff of States and of Government agencies. We do not consider Affiliates—because this non-voting category combines governments and NGOs—or Indigenous Peoples’ Organisations, because this category was established subsequent to completion of our data collection, in September 2016 (IUCN Members’ Assembly, 2016).

While institutional affiliation did not emerge as a predictor of access, nevertheless there were differences in levels of institutional access to scientific literature. Among these sectoral groups, individuals working for states and/or government agencies reported having the best institutional access (Table 5). The lowest levels of access by far are among the IUCN Secretariat, with only 28% of the staff reporting institutional access (the IUCN Library does not have an acquisitions budget). It may be that government agencies and entities are more likely than NGOs to have libraries and/or librarians to support the information needs of government workers.

Table 5 Variation by sector in proportion of respondents with institutional literature access.

Sector	Responses to Q9 (number)	Institutional access (percentage)	
IUCN Secretariat	132	28	
State and/or Government Agency Members	113	58	
International NGO Members	108	45	
National NGO Members	207	48	

How important is access to scientific literature for the conservation community?

Most respondents to our survey (regardless of institutional access) felt that easy access to scientific literature was either essential or very important to their work with IUCN (Fig. 7). This supports other findings that peer-reviewed publications remain important among science researchers generally as well as among restoration practitioners and public and private land managers (Seavy & Howell, 2010; Tenopir, Christian & Kaufamn, 2019).

Of the 1,458 respondents who felt it was either very important or essential to have easy access to scientific literature, 39% reported that they should be consulting scientific literature either sometimes (once a month) (29%) or infrequently (10%). Thus, there is a sizeable proportion of conservation professionals who do not need to access scientific literature on a frequent basis but for whom it is still very important to do so at least occasionally. For libraries with limited budgets, this could suggest that a pay-per-use model might be preferable to journal or database subscription models.

We sought to quantify the importance of online institutional access to scientific literature further by asking additional questions of those respondents who stated they did not have institutional access to scientific literature online. The majority of these respondents reported that the lack of institutional access to scientific literature online has a moderate to great negative impact on their IUCN-related work (Fig. 8). The narrative comments on this question reveal another concern beyond the negative impact on the quality of the work: time wasted trying to find appropriate literature. For example:

Figure 7 Importance of easy access to scientific literature (n = 2, 004).

Figure 8 Impact on IUCN-related work of not having institutional access to scientific literature online (n = 938).

• “I waste time searching for free versions of papers online. I waste time getting frustrated that I can’t find free versions for everything I need. I cut corners scientifically which I don’t like. I am not up to date professionally. I am not able to adequately pursue my own professional development.”

• “Time spent chasing articles from colleagues could be better spent using findings.”

The impacts of lack of access are perceived as more severe in some regions than in others. Notably, 29% of respondents from Africa reported their lack of institutional access as incurring a great negative impact; in Latin America and the Caribbean it was 24% and in all other regions <20%. Other variation was minimal: among sectors, lack of access is felt most keenly among those working for national NGOs (20% reporting great negative impact), while among disciplines it is felt most strongly by specialists in law (22%), ecosystems, and education and communication (both 21%). Among all respondents, the rate was 16%. These results can guide the efforts of funders seeking to make the greatest gains in improving access to literature for impact: for example, they suggest increased funding for conservation libraries would make particular impact within national environmental NGOs.

Most respondents reported that obtaining institutional access would have a moderate to great positive effect on the quality of their IUCN-related work (Fig. 9). Narrative comments suggest that a range of benefits would be accrued from library-facilitated access to literature online, including strengthening innovation, efficiency, and credibility:

Figure 9 Effect on quality of IUCN-related work if institutional access to scientific literature online were obtained (n = 938).

• “Work would be more thorough, more inclusive, more efficient.”

• “No effect on quality, but direct access would speed up my work at times.”

• “It will allow me to produce better Red List assessments as well as other types of reports.”

More than three-fifths of respondents without institutional access anticipate that they would access the literature frequently or very frequently if they did have access (Fig. 10)—10 percentage points higher than the 51% of all 2,285 respondents who felt that they should be accessing the literature frequently or very frequently. Thus we might expect that providing library-facilitated online access to scientific literature would allow those in the conservation community to access and use literature more frequently.

Figure 10 How frequently those reporting no institutional access would use it for IUCN-related work if they had it (n = 918).

Information pathways and preferences

We asked respondents to identify how frequently they used various means to access scientific literature; their answers shed light on the preferred (or available) pathways, both formal and informal, to scientific literature for those with and without institutional access to literature (Fig. 11).

Figure 11 Frequency of accessing literature through various information pathways among respondents with and without institutional access (n = 2, 004).

From light to dark grey: very frequently (daily); frequently (once a week); sometimes (once a month); infrequently; never or not available. (A) I use the library of my own institution. (B) I visit a local library (public, academic, etc.) to read scientific literature in print. (C) I use my institutional access to scientific literature. (D) I request journal articles from the IUCN Librarian. (E) I request journal articles from the author. (F) I access scientific literature through my own personal subscription to individual journals. (G) I ask a friend or colleague whom I know has access to scientific literature online. (H) I access whatever I can find online for free (via Google Scholar, open-access journals, ResearchGate, etc.).

Unsurprisingly, respondents with institutional access to scientific literature reported using the library of their own institution and institutional access to literature online more frequently than those without; meanwhile, those without institutional access reported asking friends or colleagues with access to literature and using free online resources (such as Google Scholar or ResearchGate) more frequently. However, accessing free online material is a popular means of accessing literature for both groups. These findings are expected, given the critical role of access in influencing information-seeking behaviour (Connaway, Dickey & Radford, 2011) and the prevalence and necessity of informal and alternative routes of access in countries with poor access to literature, such as India (Gaulé, 2009; Boudry et al., 2019). It also reflects a previous study that found open-access literature to be the most important source of information among conservation practitioners as well as university and non-university researchers in low-middle income countries (Gossa, Fisher & Milner-Gulland, 2015). Although our survey did not attempt to specifically address the impact of websites such as Sci-Hub and LibGen that enable users to download PDFs of scholarly articles, the popularity of accessing “whatever I can find online for free” among those without and with institutional access implies that such mechanisms are popular even among academic researchers (Greshake, 2016; Bohannon, 2016). Indeed, our survey might even be underreporting the popularity of accessing free papers online, given that some researchers might be prohibited from (by their institution’s firewalls) or uncomfortable with using certain sites due to their illegal nature. Nevertheless, with freely available papers obtaining 18% more citations than expected (Piwowar et al., 2018), this method of accessing literature is becoming increasingly important.

We also asked respondents about their preferred means of reading scientific literature as well as which one journal would have the largest impact on their IUCN-related work if they could obtain access to it. Together, these questions were designed to help guide strategic decision-making for conservation libraries.

Of the 2,116 respondents to the English and French surveys, 1,238 (59%) provided answers to the question “Which one scientific journal would have the largest impact on your IUCN-related work if you could obtain easy access to it?” (this question was accidentally omitted from the Spanish survey). Of these, 794 listed specific journal names, which were classified and tallied to identify those journals to which conservationists perceive that access would benefit their work most greatly. Some respondents listed more than one journal: in such cases, scores were divided among the journals listed (e.g., if four journals were listed, these were scored 0.25 each).

In total, 235 journals were mentioned by respondents, including ten listed as most desired more than ten times. These included six specialist conservation journals (Conservation Biology, Biological Conservation, Oryx, Journal of Wildlife Management, Biodiversity & Conservation, and Parks), two general science journals (Nature and Science), one general ecological journal (Ecology), and one general taxonomic journal (Zootaxa) (Table 6). There is no significant relationship between the number of times that specific journals were mentioned by respondents as those to which they most desired access and the 2015 Google Scholar h5 index value of these journals (Fig. 12). This mirrors results of weak or no relationships between popularity of journals with practitioners and their journal impact factors from conservation (Gossa, Fisher & Milner-Gulland, 2015) and other fields, such as surgery (Jones, Hanney & Buxton, 2006). Nevertheless, the variety of responses demonstrates the diversity of conservation community’s scientific literature needs, which suggests that a pay-per-view or pay-per-article model might be more cost-effective for smaller libraries than traditional journal title or database subscriptions.

Table 6 Top 20 journals mentioned by survey respondents as having the potential to have the largest impact on IUCN-related work if easy access to them could be obtained.

Rank	Journal title	
1	Conservation Biology	
2	Nature	
3	Biological Conservation	
4	Science	
5	Zootaxa	
6	Oryx	
7	Journal of Wildlife Management	
8	Biodiversity and Conservation	
9	Parks	
10	Ecology	
11	Molecular Ecology	
12	PLoS ONE	
13	Journal of Applied Ecology	
14	Zoo Biology	
15	Journal of Environmental Management	
16	Marine Mammal Science	
17	Conservation Letters	
18	Journal of Mammology	
19	Marine Ecology Progress Series	
20	Chelonian Conservation and Biology	

Figure 12 Relationship between “most desired” journals and Google Scholar h5 index of these journals (n = 235).

In addition to preferred journals, conservation professionals also have different preferred reading formats between books and journal articles. To discern this difference, we asked in question 8, “In what format do you prefer to read scientific literature?” where the choices were “I prefer reading on a screen”, “I prefer printing out to read,” and “I prefer the original hard copy.” When reading articles from scientific journals, the majority (59%) prefer reading on screen, but for books, the majority (59%) prefer to read the original hard copy. The preference for electronic journals has been noted elsewhere (Kaur, 2012).

Discussion

Our most striking findings are two-fold. First, despite the fact that 97% of respondents need it for their IUCN-related work, approximately half of the conservation community we surveyed report not having easy access to scientific literature. This stark division in ease of access to scientific literature confirms earlier findings on the difficulties of accessing literature (Cvitanovic et al., 2014; Steiner Davis et al., 2014). Second, sex, region, and, in particular, institutional access, had statistically significant effects on ease of access to scientific literature. Considering that Sci-Hub, for example, provides greater coverage than the University of Pennsylvania to “toll access” journal articles (Himmelstein et al., 2018), the persistent relevance of institutional access was surprising but nonetheless points to the need for continued support of institutional libraries.

Much concern has been raised about the challenges to the scientific process faced by Africa, Asia-Pacific, and Latin America and the Caribbean (Barber et al., 2014; Pasgaard & Strange, 2013). This geographical variation in where conservation science is produced and published is potentially related to the geographical variation in access to the literature (Karlsson, Srebotnjak & Gonzales, 2007; Fisher, 2015; Gossa, Fisher & Milner-Gulland, 2015; Nuñez et al., 2019). An information gap as well as “digital divide” (Coloma & Harris, 2005) between lower and higher income countries has long been acknowledged, and our results confirm that the conservation community in high-income countries have greater easy access than their counterparts in the rest of the world. However, even in middle-high income countries, over 40% of our respondents report not having easy access to scientific literature online. Additionally, Eastern Europe, which had the second greatest difficulty in access to the literature, is rarely highlighted in assessments of the topic. Our finding that this information gap divides sex as well as geography is presumably both a symptom and a cause of the underrepresentation of women in science (Ceci & Williams, 2011).

One approach to addressing the issue of access has been the number of worldwide programs and initiatives designed to expand scientific access to lower income countries, such as Research4Life (Burton, 2011; Bartol, 2013; http://www.research4life.org/), in which institutions in eligible countries may register for free or discounted access to scientific journals. Various individual publishers, such as the University of Chicago Press (http://www.journals.uchicago.edu/inst/ceni) and Oxford University Press (http://www.oxfordjournals.org/en/librarians/developing-countries-initiative/), offer similar programs. However, there are limitations to such systems (Smith et al., 2007; Chan, Kirsop & Arunachalam, 2011; Villafuerte-Gálvez, Curioso & Gayoso, 2007; Bendezú-Quispe et al., 2016). The factors taken into consideration to determine whether a country is eligible for Research4Life include total gross national income and the country’s Human Development Index, among others, but the combination of these factors means that no countries in Eastern Europe qualify for free access under Research4Life even though Eastern European respondents to our survey reported the second-lowest rates of institutional access to conservation literature (after Africa). Furthermore, several countries that would qualify for discounts according to World Bank criteria are not on the list (Chan, Gray & Kahn, 2012). Additionally, programmes such as Research4Life do not consider that even within high income countries, access to literature is not universal (Chan, Gray & Kahn, 2012). Finally, the Research4Life registration requires the contact information of the organisation’s Librarian or Information Specialist. However, roughly half of our survey respondents, no matter where in the world they were located, report having no institutional access to scientific literature online, which suggests the lack of an institutional library to begin with, or at best a severely underfunded one.

Although the conservation literature recognizes the research-implementation gap and even calls for investment in “knowledge brokers” (Cvitanovic et al., 2014; Sheikheldin, Krantzberg & Schaefer, 2010), it rarely acknowledges the role of libraries in improving information flow, despite the fact that access to literature is traditionally brokered by an organisation’s library. Having institutional access to literature online increases the odds of easier access to literature by nearly seven times, which suggests that core support of libraries within institutions is key to improving access. The impact of the lack of institutional access is felt not just in the quality of work being produced, but also in loss of credibility and the amount of time required to obtain papers. One study found a correlation between e-journal consumption and research outcomes (Research Information Network, 2009), suggesting that the access provided by a well-funded library could have positive impacts beyond simply saving time. Calls for evidence-based approaches in conservation that prioritize the use of synthesized knowledge such as systematic reviews over traditional journal articles, akin to those employed in medicine and public health (Pullin & Knight, 2003; Cullen et al., 2001) stop short of acknowledging the crucial role of librarians in medical systematic reviews (Harris, 2005). Even the sharing of lessons learned from field projects is impeded by the lack of institutional support to library and information management; most conservation projects fail to document their work internally, and project libraries are not well-managed (Sayer & Campbell, 2004). This suggests that donors as well as conservation institutions themselves have a role to play in supporting library and information management functions if they are truly interested in ensuring experiences and results of conservation projects are widely shared and disseminated.

Other approaches to resolving the information divide have included harnessing the growing open access movement (Laakso et al., 2011). The Budapest Open Access Initiative, which produced one of the earliest and most widely used definitions of open access in 2002, recommended two complementary strategies for achieving free and unrestricted online availability of peer-reviewed journal literature: self-archiving by authors (i.e., green open access) and open access journals (i.e., gold open access) (Budapest Open Access Initiative, 2002).

In gold open access, a paper is made immediately available for free by the publisher on the journal’s website, an approach that has been recommended in a number of influential reviews (e.g., Finch, 2013). Much of the challenge of access to the conservation literature might be resolved were funders of conservation research to require that all research outputs be published as open access (Harnad et al., 2008), a move that some major funders (e.g., US National Institutes of Health, European Union) have already adopted. Such a shift would have costs, though. Some are financial: the costs of publication is sometimes shifted from the readers to the authors, which can leave the problem of authors or their sponsoring organisations not having sufficient funds to pay the article processing charges levied by publishers for publishing in an open access journal (Siler et al., 2018; Peterson et al., 2019). One top-end estimate for how much a shift to open access would cost (for conservation science papers 2000–2013) is $51m (Fuller, Lee & Watson, 2014), funds that arguably could be better spent on conservation practice itself. However, if gold open access publishing could be shifted away from hybrid open access to full open access journals, there would be significant cost savings, since the former have been shown to be more expensive than the latter (Pinfield, Salter & Bath, 2015). Meanwhile, some publishers, like PeerJ (https://peerj.com/about/FAQ/), offer waivers to researchers from low-income countries or alternative pricing models such as author memberships. Other costs are more pernicious, such as the proliferation of “predatory publishers” (Beall, 2013).

An alternative to gold open access is green open access, whereby authors deposit post-acceptance but pre-formatting manuscripts into an online institutional or subject repository (Björk et al., 2014). Such systems have proven successful for disciplines such as physics, where arXiv respectively serves as a community-wide repository. In fact, conservation research can and has been deposited in arXiv and other preprint servers such as PeerJ Preprints, biorxiv, Zenodo, and preprints.org. The delayed and low levels of self-archiving by authors (Piwowar et al., 2018; Harnad, 2006) would still present a challenge, though.

Open access is consistent with our findings regarding information seeking behaviour: the conservation community as a whole, regardless of whether they have institutional access, turn to free material online very frequently. However, it is also not clear whether open access would save researchers time, given our finding that one of the impacts of lack of institutional access was the amount of time spent finding literature through alternate means.

Our findings emerge out of our understanding of IUCN as a broadly useful proxy for the conservation community; one example of how representative IUCN is of the conservation community is that, in 2017, there were 114 IUCN NGO Members in the USA (combined annual budget >$4.94bn) compared to 532 US NGOs (combined annual budget = $4.90bn) listed by Charity Navigator (https://www.charitynavigator.org/) in the categories “Environment”, “Wildlife Conservation”, “Zoos and Aquariums” and “Botanical Gardens” but not IUCN Members (R. Merizalde unpublished data). Nevertheless, IUCN may not be perfectly characteristic of the conservation community, and future work will require assessing the information needs of the sectors that may have been left out of our study.

In the short-term, though, our results might provide guidance to the strategic development of existing conservation libraries. Many such libraries are under severe budgetary constraints; our findings regarding conservationists’ “most desired” journals may help to guide purchasing decisions for libraries without the resources to conduct a survey of their own user’s preferred journals. In addition, our findings regarding preferred reading formats suggests that conservation libraries should continue to maintain hard copy books but could consider online-only access to scientific journals. Finally, our results should strengthen the arguments as to the importance of libraries in conservation agencies and institutions, given our strong evidence that those in the conservation community who have library-facilitated access to the literature benefit greatly in comparison to those who do not.

Conclusions

Access to scientific literature is a pernicious problem for more than half of the conservation community, with numerous negative effects as a result. Lack of institutional access is the primary predictor of disparities, followed by geographical location. In order to overcome the information divide and their subsequent limitations on conservation work, our survey results point towards solutions such as reinforcement of organisational and donor support to institutional libraries and knowledge management as well as of open access initiatives. Future work could include determining the levels of investments in libraries and information management as well as the gradations of institutional access provided by the employers (i.e., institutions) of conservation professionals, to go beyond the IUCN constituency as well as individuals’ self-reported measures of access.

Supplemental Information

Supplemental Information 1 The English-language version of the access to literature survey that was made available to respondents on SurveyMonkey

Click here for additional data file.

Supplemental Information 2 The Spanish-language version of the access to literature survey that was made available to respondents on SurveyMonkey

Click here for additional data file.

Supplemental Information 3 The French-language version of the access to literature survey that was made available to respondents on SurveyMonkey

Click here for additional data file.

Supplemental Information 4 Raw survey responses from the English, Spanish and French-language versions of the survey

Click here for additional data file.

Supplemental Information 5 The cleaned, arranged, and full-ranked dataset used to compare respondent’s answers to the demographic and professional questions on their perception of ease of access to literature

Click here for additional data file.

Supplemental Information 6 The code used to compare respondent’s answers to the demographic and professional questions on their perception of ease of access to literature

Click here for additional data file.

Supplemental Information 7 Data used to determine the most desired journals as shown in Fig. 12 and Table 6

Click here for additional data file.

We are grateful to the 2,235 people through the IUCN constituency who completed the survey; D Murith for help with survey design; A Rodrigues, R Merizalde, and K Holenstein for providing references; to G Fragoso, M Hoffmann, R Mounce, J Siikamaki, and M Lambert for comments on the manuscript; and to K Holzer, L Patterson, and the two anonymous peer reviewers for their thoughtful and constructive peer reviews.

Additional Information and Declarations

Competing Interests

Author Contributions

Human Ethics

Data Availability

Daisy Larios, Thomas M. Brooks, Nicholas B.W. McFarlane, and Sugoto Roy are all employees of the International Union for the Conservation of Nature and Natural Resources (IUCN).

Daisy Larios conceived and designed the experiments, performed the experiments, analyzed the data, prepared figures and/or tables, authored or reviewed drafts of the paper, and approved the final draft.

Thomas M. Brooks conceived and designed the experiments, performed the experiments, analyzed the data, authored or reviewed drafts of the paper, and approved the final draft.

Nicholas B.W. Macfarlane and Sugoto Roy analyzed the data, authored or reviewed drafts of the paper, and approved the final draft.

The following information was supplied relating to ethical approvals (i.e., approving body and any reference numbers):

IUCN has no IRB or equivalent mechanism: approval for the research was provided through supervisory channels within the institution’s Science and Knowledge Unit. The survey on which data collection was based was wholly voluntary. The final question gave respondents the option of providing their name and email address, in case any follow-up was necessary, but all responses to this question were deleted from the dataset before submission to PeerJ such that no response can be individually identified.

The following information was supplied regarding data availability:

The raw data and code are available in the Supplementary Files.

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
