# Peer review of "Access to scientific literature by the conservation community"

_PeerJ, doi:10.7717/peerj.9404_

## Round 0.1 · original submission · Major Revisions

Four reviewers have no assessed your manuscript and find the study to be generally well done. However, the reviewers have a number of comments that will help clarify your results to the broad readership of this study. I encourage you to pay particular attention to Reviewer 2's comment about the survey only being done on the IUCN, Reviewer 3's comment about the ambiguity about the regression, and Reviewer 4's comment about the illegality of some of the ways research is freely accessed.

I provide some of my own comments below as well. I look forward to a revised version of this manuscript.

My comments:

Your discussion opens up by saying half of the conservation community does not have access to the scientific literature. The more accurate statement is that half of the IUCN community doesn't have this access. The reviewers have noted a similar issue and it's worth clarifying your arguments throughout the manuscript that your respondents are only the IUCN community. An interesting discussion point to highlight for future work is the extent to which the IUCN community is representative of the conservation community broadly and whether it has similar or disproportionately high / low access relative to other conservation entities.

Please rephrase developed and developing countries to "low income", "low-middle income", "middle income", and "high income" countries instead. Or use a similar alternative that is more appropriate and which doesn't carry the same colonial connotation.

Line 358-359: it's unclear if / what statistical analysis was used here. It looks like a simple linear regression but the analysis and results (p-values, R2, etc) should be reported in the methods and results. Critically, when looking at the figure, the authors have flipped the dependent and independent variable. The google scholar index should be the independent variable (X-axis) and the number of mentions should be the dependent or response variable (y-axis). Given the response variable is the number of mentions - and is therefore count data - the authors should be using a Poisson or negative-binomial generalized linear model (GLM) to analyze the data. Please address this since this is an important analysis for this study. This point could also be a larger discussion point in that it might help conservation researchers target particular journals more if it's known that practitioners are more keen on particular journals.

Table 1: for your logistic regression (binomial) model, please use a Tukey's post hoc test (in R this can be done with the multcomp package) to conduct pairwise comparisons among the geographic regions. This would provide interesting context for the relative rankings among regions. I understand that now your contrasts are just set up to compare all other regions to Africa. However, all we learn from this analysis is the Western Europe and Africa are different. But it would be useful to understand whether other geographic regions are equivalent or are also different in some ways (e.g., are Eastern and Western Europe the same).

You should also include the odds ratios in your table. You present coefficients but you discuss odds ratios (or at least that's what I presume you discuss) in the results and it would be helpful to see those ratios in the table.

Additionally, for the analysis shown in Table 1, it would be nice if the authors explored interaction terms between sex and institutional access, sex and geography, and geography and institutional axis. There may be some interesting non-additive patterns that come from these data.

Another analytical layer you might consider adding is whether GDP or some other national metric for income is related to access. Right now you've clustered your geographic information into regions but perhaps there are some other geopolitical or economic variables that might tease apart some of the underlying issues. This could be particularly interesting given your discussion contrasting Eastern Europe and regions like Africa with similar levels of access.

Figure one is not particularly helpful. Please consider removing this and just stating the results in the text.

Figure 2: please reorganize the legend and pie slices so the pieces are in a logical order (i.e., very frequently to never).

A general figure comment is to change the color differences. Some of your figures rely on contrasting green and red which is hard on color-blind readers. A different color palette would be preferable and a number of sources online provide palettes that are broadly suitable. Additionally, the color palette changes a bit from figure to figure - consider choosing a consistent scheme to make it easier on the readers to digest the data.

You have a nice discussion about the importance and challenge of open access. One additional discussion point you may wish to consider making is that journals like PeerJ offer one "no questions asked fee waiver" to researchers from low-income countries each year. Additionally, PeerJ works to create institutional plans so that researchers associated with various institutions can publish one paper for free each year. This latter approach certainly isn't possible for all types of institutions but it may help in some cases where conservation entities may have enough funding to create a plan like this.

Reviewer 1 ·

Basic reporting

No comment

Experimental design

No Comment

Validity of the findings

No Comment

Additional comments

Line 21: ~50% of the 97% or of the whole?
Line 31: Change "about" to "roughly"
Line 64: "academics AND consultants"

·

Basic reporting

No comment

Experimental design

No comment

Validity of the findings

No comment

Additional comments

The manuscript “Access to scientific literature by the conservation community” describes survey results regarding the needs, methods, and barriers to access to scientific literature by the constituency of the International Union for the Conservation of Nature. It provides a quantitative assessment for an issue which is often cited but rarely assessed systematically. It found that the vast majority the constituency feel that the literature is important to their conservation work, and that about half have substantial barriers to accessing it. The studied showed that a major factor was institutional access to the literature.

The study is well-written and driven by clear research questions. The analyses are sound, and the conclusions are based on the findings. I feel that the discussion could be improved by clarifying the publishing and access options. The study may also have limitations in that it focused only on IUCN members. I discuss these aspects below along with other minor inline suggestions.

L30-32: They way the abstract is written, it could be misconstrued that this important conclusion is inferred from the 55% with access rather than direct response to a question asking about this challenge. Maybe this could be reworded to make that explicit, like in L165-167.

L53-54: Inclusion of the entire IUCN constituency is a strength of this study. There are also many conservation professionals who conduct on-the-ground conservation actions who would generally not be included here. This is especially true of small organizations and/or those whose focus is not conservation, but who have staff members who are dedicated to this work (e.g. watershed councils and city governments). It would be difficult to reach all of these individuals, so it is understandable that they were not included here. However, it would be good to mention that they are not captured and are probably even less likely to have institutional access to literature than many IUCN members.

L157: Extra “had”.

L169-186: Would this work better in the Methods section?

L214-218: It is counterintuitive that 52% is significantly different than 54% here. Is this a case where the sample size is so high that you might be getting statistically significant results which aren’t necessarily biologically significant? I appreciate reporting the factors so that we can see the overall impact, but maybe this should be discussed.

L354-355: I think the result of there being 235 journals listed out of 794 responses is in itself important. I think it speaks to the fact that most conservation practitioners often need very specific information relating to their taxon/biome/threat of focus. I think this demonstrates why buying access to a few big journals and hoping that many people will use them will likely miss the mark for most conservationists. It may be more cost effective to have a budget to pay per article where open access is lacking. I think it would be useful to stick in a sentence about the fact that so many journals were mentioned as a demonstration of this needed variety.

L364-367: This paragraph seems to be standing alone. Where are the associated questions and numbers?

L369: At some point it seems relevant to state that institutional access is a spectrum rather than a binary, and that some institutions may provide access to some journals and not others. Some of this is lost when asking the binary question of whether they have institutional access or not, and may explain why those “with” access still use other means to obtain literature.

L369: In general, I think that the discussion could benefit from a short, basic introduction to the various forms of access and options of publishing. Although it is touched on, I feel that practitioners (including myself) are often baffled by the whole process. As you mentioned, people are unfamiliar with what a “library” is, and I know that among my colleagues we are constantly confused about where publication/access fees go, what “open access” is, and just the general model of publication and access. I think it would be great if you had a short paragraph which laid these things out before getting into the specifics.

L439-441: I have not heard of “gold” (or “green”) open access before, and I’m not sure that this sentence clarified it for me.

L450-452: This sentence seems important, but I am not understanding what it means or who would be saving money.

L470: It looks like the rest of this sentence is missing.

L475-478: This is a good, clear conclusion.

Fig. 1: Spell out the languages on the figure.

Fig. 2: For all figures, I would suggest ordering the responses from most positive to most negative in a clockwise fashion starting from the top. I appreciate that the color coding is in order, but skipping around the circle is a bit confusing.

Figs. 3 and 4: These are really important results and are presented clearly.

Fig. 12: I had to stare at these for a while to find the take-home messages, but, once I did, I found them to be important. It is a matter of taste, but I might suggest flipping the ordering of the stacks (so that daily is on the bottom), getting rid of the grey (so that the bars represent responses of access), and not making the “infrequently” stand out from the others so much in color, since they are all gradients of a positive. For me, these changes would help me to quickly see that most methods don’t differ much between those with and without access (other than using the institutional access), and that most people look online for free.

Fig. 12: The result that most people get what they can online for free seems like a major finding of this study. I think it should be in the abstract.

Reviewer 3 ·

Basic reporting

I would suggest that the history of the IUCN is not needed - it doesn't add much to the main message of the paper. Instead, I would replace this information with more background on the previous work that has been done to look into lack of access to scientific papers and journals globally. There is a body of literature on this topic that the paper touches on briefly, but the introduction could benefit from a more in-depth look at this.

The authors should explain the response rate for the survey and establish representativeness, especially given the reported low number of responses in Spanish and French.

The last sentence of the conclusion seems to cut off. If indeed this is not simply an uploading error, the authors need to finish their last sentence.

Experimental design

The authors should include a definition of “constituency” for the reader. As it currently stands, it is somewhat hard to determine who was actually surveyed by the authors. Consider including a figure explaining the structure of the IUCN and its constituents for clarity.

Validity of the findings

Given the nature of this study (a surveying method), the authors should consider phrasing the results as “participants reported that…” rather than stating the results as if they are inherently correct. It’s quite possible that the survey results reflect the actual situation at hand, but changing the phrasing allows for a more nuanced view given that results stem from survey responses.

The authors should report in the text that while “region” was a statistically significant predictor when it was Western Europe/US/Canada/Australia/New Zealand, the other regions were not significant predictor levels.

The authors don’t give much information on how they did the regression in Figure 13, which makes it feel a bit abrupt. The authors also don’t report a p-value. Additionally, “perceived desirability” is an unclear way of speaking about this survey question. Does it mean participants only thought they desired the journal? Does the “perceived” qualifier pertain to the authors or the survey participants? This regression could use more clarification.

Additional comments

The authors should condense the number of figures within the article. The figures are useful but having 13 individual figures is unwieldy. Perhaps some figures could be condensed into a multi-figure panel, or some of the figures could be moved to the supplemental information?

It would be helpful to clarify what constitutes a "library" earlier in the article. The authors provide more of a background the concept of a “library” when describing their survey methods, but they introduce the concept of libraries earlier in the introduction without defining it.

·

Basic reporting

No comment

Experimental design

No comment

Validity of the findings

The first sentence in the Conclusions section on Lines 473-474 is the main reason this topic is so important. I think the paper would benefit from including some examples of the "numerous negative effects" that lack or difficulty of access to scientific literature has had on conservation practitioners' ability to carry out their work.

Additional comments

The paper is well written, clear, concise, and important. Conclusions are adequately and appropriately drawn from the results. However, the authors' surprise at the persistent relevace of institutional access (Lines 375-378) given the availability of free downloads on SciHub and LibGen does not take into account that the material on these sites is pirated, and therefore illegal. Some governmental organizations may preclude their employees from using sites like that by blocking internet access, and some people may have an ethical problem with breaking the law. I think including the legal status of those sites is relevant to why there is still such a reliance on institutional access and should be disclosed in the paragraph where they are referenced (Line 334).
In addition, I have three editorial comments:
Line 157: omit the word "had" betweeen "have" and "internet"
Lines 399-400: ensure the hyperlink is activated like the others
Line 471: text is missing from what was likely the last sentence of the paragraph

---

## Round 0.2 · Minor Revisions

Thank you for your thoughtful and comprehensive revision. The manuscript is greatly improved and I have only a small number of minor revisions that need to be made.

First, I appreciate the authors making clear the IUCN community broadly is a representative proxy for the conservation more broadly. That being said, it is absolutely critical to ensure you do not overemphasize this. This study of yours is a phenomenal contribution and will hopefully motivate more work to better untangle how different members of the conservation community are able to access scientific literature. However, it's critical to circle back in the discussion and mention that while the IUCN is a broadly useful proxy of the conservation community, it is not perfectly reflective and future work will need to understand what sectors of the community may be left out of this survey.

Second, it is absolutely critical - and in accordance with PeerJ's instructions - that you publish the data associated with Figure 12. There are some really interesting results in this table that would benefit from those data being published. I'd even strongly encourage you to maybe take the top 20 or 30 results and put those in a table in the main text (and reference the full results in the supplement). These top results would include open access journals like PLoS ONE and journals like Molecular Ecology that are typically considered less applied.

Third, In your rebuttal you say you can’t figure out the response rate. But you also argue that IUCN is a useful proxy for the conservation community broadly because it’s constituency generally reflects the conservation community at large. However, it’s important to know whether those who responded to you survey actually reflect the broader community. If we assume, for instance that your respondents are a biased subset of the IUCN, the data here could either be a conservative estimate of access to the literature or the opposite.

Line 75: please clarify in the parenthetical that these examples “whose focus may not explicitly be conservation”. As it’s written now it sounds like these professional aren’t doing conservation but in reality they very much are even though their EMPLOYERS may not be conservation-specific.

Please provide not only the raw data but also the data you analyzed. As per PeerJ’s policies, please also provide the code you used to analyze these data.

Line 222-224 and elsewhere: provide the statistics or reference where the statistics for these claims are

Line 235-238: we need to know all the pairwise differences that are significant. Not just “others”. This can be a table or completely laid out in the results. As written, it’s not clear if Africa is just worse than everywhere else or Western Europe and a subset of other regions. I think a table with Tukey’s coefficients and p-values would be extremely valuable.

With respect to your revision about sites like SciHub, I highly recommend being more explicit by stating these databases aren’t just illegal but are inaccessible to some practitioners because they are illegal. This was the point of the reviewer’s comment. Government employees may not be able to access them due to firewalls. Please revise this section accordingly.

This is a well done paper and I look forward to your revision.

---

## Round 0.3 · accepted · Accept

Thank you for your thorough revision. This is a rich manuscript that will be immediately useful for conservation entities globally. I very much appreciate all the work the authors have done not only to compile these data and write the results but also in revising this work to greatly improve it. Congratulations on an excellent study.

Contingent upon acceptance is the change in a single word. I forgot to include in my last message to change the word "gender" to "sex" throughout the text, including the Table. Gender reflect a lot more complexity than what is shown in the categorical options you provided and "sex" is the more accurate - albeit still imperfect - term.

I look forward to the publication of your study, congratulations.